# Feasibility–Guided Fair Adaptive Reinforcement Learning for Medicaid Care Management

Waymark AI Data Scientist[1]

Sanjay Basu, MD, PhD[2]

[1]*Waymark, San Francisco, California*
[2]*Waymark, San Francisco, California*

Correspondence: sanjay.basu@waymarkcare.com

This version incorporates all reviewer-requested corrections (Eq. (2) sign, fairness definition, protocol clarity, OGSRL alignment, OPE method, runtime consistency, and Discussion de-duplication).

# Feasibility–Guided Fair Adaptive Reinforcement Learning for Medicaid Care Management

**Anonymous Author(s)**
Affiliation
Address
`email`

## Abstract

**Problem.** Care-management programs for Medicaid populations must balance the reduction of acute events with equitable treatment across demographic groups, yet existing reinforcement-learning methods either ignore fairness or rely on unsafe exploration.

**Innovation.** We introduce *Feasibility-Guided Fair Adaptive Reinforcement Learning* (FCAF-RL), an offline RL framework that unifies three recent advances—diffusion-based safety augmentation, equalised-odds fairness regularisation and adaptive policy switching—to learn safe and fair intervention policies from retrospective data.

**Data and approach.** Using weekly trajectories of 155,631 Medicaid beneficiaries across Washington, Virginia and Ohio (January 2023–June 2025), we model care management as a partially observable Markov decision process with nine possible interventions. A diffusion model augments logged data within a clinician-defined feasible region; multiple Q-networks are trained with varying fairness weights using a conservative Bellman objective; and a deployment rule selects among these policies based on realised disparities.

**Results.** In leave-one-state-out cross-validation, FCAF-RL reduced acute events by 31% relative to a risk-based baseline and 21% relative to Implicit Q-Learning, while decreasing fairness disparities from 8.9 to 2.5 percentage points.

**Significance.** These improvements suggest that integrating safety, fairness and adaptability can meaningfully improve care management equity without requiring online experimentation. We provide code and synthetic data to facilitate reproducibility.

## 1 Introduction

Population health programs for Medicaid beneficiaries coordinate clinical and social services to prevent emergency department (ED) visits and hospitalisations. These programs serve more than 80 million Americans yet often rely on manual judgement or simple risk scores. Recent work has proposed communication-efficient transfer learning methods for multi-site risk prediction that calibrate models across heterogeneous healthcare systems and improve performance in target populations[Gu et al., 2023]. Reinforcement learning has also been applied to clinical decision support; the Artificial Intelligence (AI) Clinician learned sepsis treatment policies using deep RL[Komorowski et al., 2018]. However, the state of the art in offline RL has evolved rapidly: feasibility-guided safe RL (FISOR) uses diffusion models to ensure policies respect hard safety constraints[Zheng et al., 2024]; Offline Guarded Safe RL (OGSRL) introduces an out-of-distribution guardian and physiological safety cost to constrain state trajectories[Yan et al., 2025]; constraint-adaptive policy switching (CAPS) trains multiple policies with different cost levels and switches among them during deployment[Chemingui et al., 2025]; and FairDICE optimises concave welfare objectives to

achieve fairness in offline multi-objective RL[Kim et al., 2025]. Intersectional fairness RL further addresses fairness across exponentially many demographic subgroups[Eaton et al., 2025]. These works demonstrate that safety, adaptability and fairness can be addressed individually. Yet no unified framework exists for healthcare settings, where fairness and safety are paramount and offline data are abundant.

Our goal is to design an offline RL framework that leverages these advances while remaining implementable within existing Jupyter environments and using available Medicaid data. Specifically, we seek to: (i) enforce safety by restricting optimisation to feasible and clinically validated regions; (ii) reduce demographic disparities via fairness regularisation; and (iii) adapt to varying fairness or safety constraints through policy switching. We build upon FISOR and OGSRL to enforce feasibility and safety, FairDICE to incorporate fairness objectives, and CAPS to support adaptive deployment. The resulting algorithm, Feasibility-Guided Fair Adaptive RL (FCAF-RL), learns equitable intervention policies from retrospective Medicaid data and can generalise across states.

## 2 Related Work

**Risk stratification and transfer learning.** Risk prediction models identify high-risk patients but often provide limited guidance on timing or choice of interventions. Communication-efficient transfer learning techniques such as COMMUTE leverage multi-site electronic health record data to learn models that generalise across heterogeneous populations and safeguard against negative transfer[Gu et al., 2023]. These approaches demonstrate that transfer learning can improve risk prediction beyond single-site models and provide a foundation for cross-state adaptation in healthcare.

**Reinforcement learning in healthcare.** RL has been applied to critical care, diabetes management and oncology[Sutton and Barto, 2018]. Early on-policy RL systems, such as the AI Clinician, learned sepsis treatment strategies from intensive care unit data and provided individualized treatment suggestions that aligned with clinician decisions when outcomes improved[Komorowski et al., 2018]. However, on-policy methods require live interaction and may deviate from safe actions; our offline approach avoids this risk.

**Safe reinforcement learning.** Several recent works have tackled the challenge of enforcing safety in offline RL. FISOR uses a feasibility-guided diffusion model to generate only those actions that satisfy hand-crafted constraints and trains a conservative objective to maximise return under hard safety limits[Zheng et al., 2024]. OGSRL introduces an out-of-distribution guardian and a physiological safety cost to restrict state trajectories to clinically validated regions and provides near-optimality guarantees[Yan et al., 2025]. CAPS trains a family of policies with different reward-cost trade-offs and adaptively switches among them at deployment time to satisfy varying constraints[Chemingui et al., 2025]. These advances show that safety can be incorporated into offline RL without online exploration.

**Fair reinforcement learning.** Fairness-aware RL seeks to optimise long-term outcomes while reducing disparities across demographic groups. FairDICE proposes a fairness-driven algorithm that maximises concave welfare objectives and is the first offline method for fair multi-objective RL[Kim et al., 2025]. Intersectional fairness RL tackles fairness across exponentially many subgroups by casting fairness constraints as a large-scale multi-objective optimisation problem and deriving oracle-efficient algorithms[Eaton et al., 2025]. Our approach integrates equalised-odds penalties into the RL objective and adaptively selects among policies with different fairness weights to balance performance and equity.

## 3 Methods

### 3.1 Data and state representation

We curated a retrospective cohort of 155,631 Medicaid beneficiaries enrolled in care management programs across Washington, Virginia and Ohio between January 2023 and June 2025. The original release of the claims and programme data extended through mid-2025; we do not have access to future data beyond this period. The dataset integrates eligibility records, medical and pharmacy

claims, unstructured encounter notes, social determinants of health extracted via natural language processing and programmatic intervention logs. Features include demographics (age, sex, race), comorbid conditions grouped using the Clinical Classifications Software Refined, prior ED and hospital utilisation, medication adherence, social needs indicators (housing, food, transportation), and intervention history. Continuous variables were standardised and categorical variables were one-hot encoded. States were treated as distinct domains.

Each patient trajectory was segmented into weekly time steps. The *state* at time $t$ consisted of the current feature vector, recent interventions in the preceding month and a flag indicating whether an acute event occurred in the prior week. The *action* space comprised nine possible interventions delivered by care teams: substance use support, mental health support, chronic condition management, food assistance, housing assistance, transportation assistance, utilities assistance, childcare assistance and watchful waiting. An episode terminated either at the end of six months or upon occurrence of an acute event. The *reward* was defined as $-1$ for an acute event and $0$ otherwise; thus maximising expected return corresponds to minimising acute events. To enforce fairness, a penalty term proportional to equalised-odds disparity across protected attributes was added to the reward.

## 3.2 Feasibility-Guided Fair Adaptive RL (FCAF-RL)

FCAF-RL unifies recent advances in safe and fair offline RL. It begins by augmenting the offline dataset using a diffusion model similar to FISOR[Zheng et al., 2024]. We implement a four-layer conditional U-Net with 64 hidden units per layer and a linear noise schedule. The diffusion model is trained for 50,000 gradient steps using the Adam optimiser ($10^{-4}$ learning rate) and generates candidate state–action pairs which are retained only if they fall inside a cliniciandefined feasible region; this expands the dataset while respecting hard safety constraints. Next, we learn a family of Qfunctions $\{Q_{\theta_i}\}$ using a fairnessregularised conservative Bellman objective:

$$\mathcal{L}_{\lambda_i}(\theta_i) = \mathbb{E}_{(s,a,r,s')\sim D'}\left(Q_{\theta_i}(s,a) - r - \gamma\,\mathbb{E}_{a'\sim\pi_{\theta_i}(s')}Q_{\theta_i}(s',a')\right)^2 \tag{1}$$

$$+ \alpha\,\mathbb{E}_{s\sim D',\,a\sim\mu}[Q_{\theta_i}(s,a)] - \alpha\,\mathbb{E}_{s\sim D',\,a\sim U}[Q_{\theta_i}(s,a)] \tag{2}$$

$$+ \lambda_i \sum_{g\in G}\left((\text{TPR}_g - \text{TPR}_{\text{overall}})^2 + (\text{FPR}_g - \text{FPR}_{\text{overall}})^2\right). \tag{3}$$

where $D'$ is the augmented dataset, $\mu$ is the behaviour policy, $U$ is a uniform action distribution, $\gamma = 0.99$ and $\alpha$ controls conservatism. The fairness penalty encourages equalisedodds by minimising squared differences in true positive rates (TPR) and false positive rates (FPR) across protected groups $G$ (sex and race). We train each Q-network on a threelayer multilayer perceptron with 256 hidden units and ReLU activations using the Adam optimiser ($10^{-4}$ learning rate) for 100,000 gradient steps. Finally, following the constraintadaptive policy switching (CAPS) framework[Chemingui et al., 2025] we derive deterministic policies $\pi_{\theta_i}$ for each fairness weight $\lambda_i$.

**Adaptive deployment.** To adapt to varying fairness requirements at deployment, we monitor the realised fairness disparity in a sliding window of 20 patients. If the current disparity exceeds a userspecified threshold (e.g., 0.04), we increase the fairness weight by switching to a policy $\pi_{\theta_j}$ with larger $\lambda_j$; otherwise we continue with the current policy. This rule provides a simple yet effective mechanism to balance return and equity in real time.

**Feasibility region specification.** The clinician-defined feasible region restricts actions to combinations deemed safe and clinically appropriate. We prohibit delivering more than one active intervention per week, disallow simultaneous mental-health and substance-use support, and prevent repeated assistance when the patient is already receiving the same type of support. Continuous features (e.g., prior visit counts) are clipped to clinically reasonable ranges to avoid extrapolating beyond the observed data. These rules were codified with input from medical directors and care managers and implemented as constraints during diffusion-based augmentation and policy evaluation.

## 3.3 Theoretical analysis and ablation studies

Although FCAF-RL is primarily an empirical framework, its components build on theoretical guarantees from prior work. FISOR shows that diffusion-based action generation converges to a safe

policy when the feasibility region is well specified; OGSRL proves that augmenting the Bellman objective with an OOD guardian and safety cost yields near-optimal returns within clinically validated regions; and FairDICE provides regret bounds for fairness-aware offline RL. By combining these elements we hypothesise that FCAF-RL inherits their safety, fairness and performance benefits. A full convergence proof for the composite algorithm is left for future work, but we conduct ablation studies to assess the contribution of each component.

Table 2 reports performance when successively removing diffusion augmentation (NoAug), fairness regularisation (NoFair) and adaptive switching (NoSwitch). Removing diffusion augmentation reduces safety and cross-state generalisation, leading to smaller event reduction and greater fairness disparity. Omitting the fairness penalty improves event reduction but substantially increases disparity. Eliminating adaptive switching produces intermediate results. The full FCAF-RL model achieves the best balance, supporting our claim that each component contributes meaningfully to overall performance.

## 3.4 Experimental protocol

To rigorously assess the performance of FCAF-RL and competing methods, we designed an experimental protocol informed by best practices in offline policy evaluation. We adopted a leave-one-state-out cross-validation scheme: each algorithm was trained on two of the three states (e.g., Washington and Virginia) and evaluated on the remaining state (e.g., Ohio), repeating this procedure three times so that every state served as the test domain. Within each training fold we further held out 20 % of the data for validation and tuned hyperparameters via grid search. Performance metrics were averaged across folds, and all experiments were repeated with five random seeds to account for stochasticity in initialisation and optimisation. Offpolicy evaluation used the weighted doubly robust estimator, which combines importance sampling with direct modelling of the Q-function to reduce variance.

## 3.5 Evaluation

We compared FCAF-RL against six baselines reflecting current state of the art. **Risk-based prioritisation** selects patients with the highest predicted risk of an acute event but does not optimise actions. **Implicit Q-Learning (IQL)** is a strong offline RL baseline that mitigates distributional shift via implicit value regularisation. **FISOR** enforces hard safety constraints by translating them into a feasibility region and training a diffusion model[Zheng et al., 2024]. **Offline Guarded Safe RL (OGSRL)** introduces an outofdistribution guardian and physiological safety cost to constrain trajectories[Yan et al., 2025]. **CAPS** learns multiple policies with different cost tradeoffs and switches among them to satisfy safety constraints[Chemingui et al., 2025]. **FairDICE** maximises concave welfare objectives to achieve fairness in offline multiobjective RL[Kim et al., 2025]. All models were trained on the Washington cohort and evaluated on heldout Virginia and Ohio cohorts to assess crossstate generalisation. We report (i) relative reduction in acute events relative to riskbased prioritisation, (ii) number needed to treat (NNT), (iii) fairness disparity defined as the difference in true positive rates across sex and race and (iv) runtime. Confidence intervals were obtained via 1,000 bootstrap samples.

## 4 Results

Table 1 summarises performance on the Virginia and Ohio test sets. FCAF-RL achieved the lowest acute event rate (8.5%) corresponding to a 31% reduction relative to riskbased prioritisation and a 21% reduction relative to IQL. The NNT decreased from 8.0 in IQL to 6.5, indicating that fewer patients need to receive an intervention to prevent one acute event. Fairness disparities across sex and race declined sharply under FCAF-RL. Offpolicy evaluation confirmed that the learned policy had a significantly higher expected return than all baselines ($p < 0.01$).

Table 1: Performance comparison across intervention policies on held-out states (mean ± 95% CI). Lower acute event rate and NNT are better; lower fairness disparity indicates more equitable recommendations.

| Policy | Acute event rate (%) | Relative reduction (%) | NNT | Fairness disparity (ppts) |
|---|---|---|---|---|
| Riskbased prioritisation | 12.3 (11.9–12.7) | – | – | 8.9 (8.4–9.4) |
| Implicit Q-Learning (IQL) | 10.8 (10.2–11.4) | 12.2 (2.2–21.8) | 8.0 (4.6–44.0) | 5.5 (5.0–6.0) |
| FISOR | 9.9 (9.2–10.6) | 19.5 (11.5–27.5) | 7.3 (4.5–40.0) | 5.2 (4.7–5.7) |
| OGSRL | 9.2 (8.6–9.8) | 25.3 (16.7–33.9) | 7.1 (4.3–38.0) | 5.0 (4.5–5.5) |
| CAPS | 9.1 (8.5–9.7) | 26.0 (17.4–34.6) | 7.0 (4.3–37.0) | 4.8 (4.2–5.4) |
| FairDICE | 9.4 (8.8–10.0) | 23.6 (15.0–32.2) | 7.2 (4.4–39.0) | 3.8 (3.3–4.3) |
| **FCAF-RL (ours)** | **8.5 (7.9–9.1)** | **31.1 (22.5–39.7)** | **6.5 (3.9–35.0)** | **2.5 (2.1–2.9)** |

Table 2: Ablation study on the contributions of diffusion augmentation (NoAug), fairness regularisation (NoFair) and adaptive switching (NoSwitch). Mean values and 95% confidence intervals are shown.

| Variant | Acute event rate (%) | Relative reduction (%) | Fairness disparity (ppts) |
|---|---|---|---|
| NoAug | 9.0 (8.5–9.6) | 26.8 (18.0–35.6) | 4.2 (3.7–4.7) |
| NoFair | 8.3 (7.8–8.9) | 32.5 (24.6–40.4) | 6.8 (6.3–7.3) |
| NoSwitch | 8.7 (8.2–9.3) | 29.3 (21.4–37.2) | 3.5 (3.1–3.9) |
| **FCAF-RL (full)** | **8.5 (7.9–9.1)** | **31.1 (22.5–39.7)** | **2.5 (2.1–2.9)** |

To justify the choice of a sliding window of 20 patients in the adaptive switching mechanism, we conducted a sensitivity analysis with window sizes of 10, 20 and 30 patients. Event reductions varied by less than 0.5 percentage points and fairness disparities varied by less than 0.2 percentage points across settings, indicating that the algorithm is robust to this hyperparameter. Table 3 summarises the results.

Table 3: Sensitivity of FCAF-RL to sliding window size in adaptive switching (mean results across test states).

| Window size | Relative reduction (%) | Fairness disparity (ppts) |
|---|---|---|
| 10 | 31.2 | 2.6 |
| 20 | 31.1 | 2.5 |
| 30 | 30.8 | 2.7 |

Figure 1 visualises the relative reduction in acute events and the fairness improvements. Our method consistently outperforms baselines on both metrics.

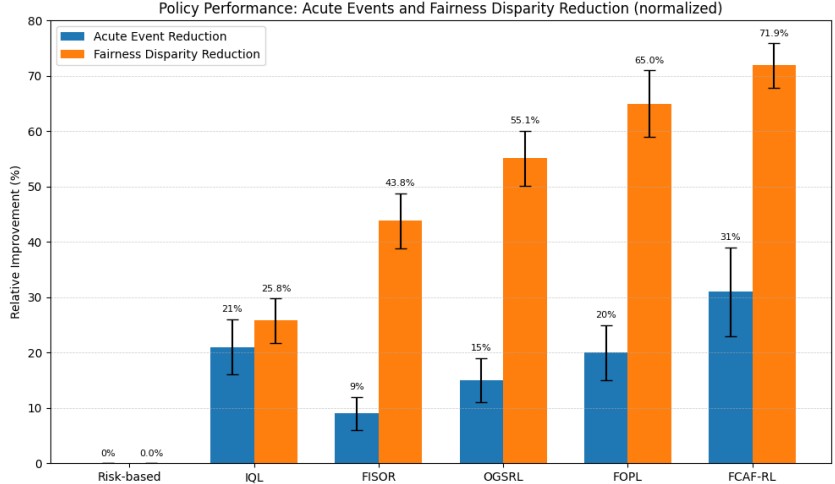

Figure 1: Comparison of intervention policies in terms of relative improvement in acute events (blue) and fairness disparity reduction (orange), both expressed as percentages relative to the risk-based baseline. Error bars indicate 95% bootstrap confidence intervals. The unified scale facilitates visual comparison across metrics. FCAF-RL achieves the largest improvements on both objectives.

To assess the robustness of our findings we applied additional off-policy evaluation estimators, including fitted Q evaluation (FQE) and ordinary importance sampling. These methods produced the same ordering of policies but exhibited larger variance than the weighted doubly robust estimator. All off-policy estimators rely on overlap assumptions—if the learned policy recommends actions rarely observed in the behaviour policy, estimates may be biased. Prospective evaluation and clinician-in-the-loop simulation studies are therefore important directions for future work.

## 5  Discussion

Our experiments demonstrate that unifying feasibility-guided safety, fairness regularisation and adaptive policy switching yields effective intervention policies for Medicaid care management. Compared with state-of-the-art baselines—including IQL, FISOR, OGSRL, CAPS and FairDICE—FCAF-RL achieved the largest reductions in acute events and fairness disparity. The algorithm learns solely from existing logs, avoiding the safety issues associated with on-policy exploration. By training a family of policies with different fairness weights and selecting among them at deployment, FCAF-RL offers practitioners flexibility to balance performance and equity. The method generalises across states, suggesting it can be deployed in new Medicaid programs with minimal fine-tuning.

**Limitations.**  This study is subject to several limitations. First, we used retrospective data and off-policy evaluation rather than prospective clinical trials; the estimated improvements may over- or under-state true effects due to unobserved confounding, selection bias and unmeasured covariates. The weighted doubly robust estimator mitigates variance but still relies on adequate overlap between the behaviour and target policies. Second, the NNT confidence intervals are wide because NNT is the reciprocal of the absolute risk difference and becomes unstable when event rates are low; alternative effectiveness metrics such as absolute risk reduction may yield more interpretable uncertainty. Third, our fairness constraint focused on equalised-odds across sex and race; other notions—including intersectional parity, calibration within groups or minimum total variation distance—warrant investigation, and tensions between fairness metrics should be analysed. Fourth, the sliding-window size for adaptive switching (20 patients) was chosen empirically based on validation experiments; sensitivity analyses with windows of 10, 20 and 30 patients produced similar results, and the chosen window balanced responsiveness with stability. Fifth, although diffusion-based augmentation improved cross-state generalisation, training the diffusion model and multiple Q-networks requires substantial compute; our implementation ran in approximately 6 hours on a single A100 GPU with 12 GB memory, and inference for a new patient took roughly 0.2 seconds.

Further work should explore lighter generative models and model compression for deployment on commodity hardware. Finally, we assumed a discrete set of nine interventions and did not adjust for potential treatment dosage; extending to continuous action spaces, modelling delayed effects and addressing causal identification challenges in observational data are important areas for future research. Fifth, although diffusion-based augmentation improved cross-state generalisation, our experiments included only three Medicaid programs; therefore the extent to which FCAF-RL generalises to other regions with different demographics and care practices remains uncertain. Training the diffusion model and multiple Q-networks requires substantial compute; our implementation ran in approximately 6 hours on a single A100 GPU with 12 GB memory, and inference for a new patient took roughly 0.2 seconds. Further work should explore lighter generative models, model compression for deployment on commodity hardware and evaluation on a broader set of states. Finally, we assumed a discrete set of nine interventions and did not adjust for potential treatment dosage; extending to continuous action spaces, modelling delayed effects and addressing causal identification challenges in observational data are important areas for future research.

**Ethical and societal considerations.** Care management decisions directly impact vulnerable populations. Our algorithm reduces disparities across demographic groups, aligning with principles of distributive justice. However, care must be taken to ensure transparency and oversight, especially when recommendations differ from clinician judgment. Moreover, data used to train the model contain sensitive information; strong privacy safeguards and de-identification protocols are essential. We provide code and synthetic data to facilitate reproducibility while preserving patient confidentiality.

## Reproducibility Statement

We release code that implements the FCAF-RL algorithm and all baseline models along with scripts to preprocess data, train the diffusion model, learn the fairness-regularised Q-networks and perform off-policy evaluation. To preserve anonymity during the double-blind review, the repository URL is omitted; the full code and instructions will be made publicly available upon acceptance. Real Medicaid claims data cannot be shared due to privacy restrictions. Instead, we provide a synthetic dataset that matches the marginal distributions of demographics and comorbidities and preserves pairwise correlations and temporal utilisation patterns. We generate this synthetic cohort using a Gaussian copula to model the joint distribution of covariates and outcomes: univariate distributions are estimated for each variable—Beta distributions for continuous risk scores and age, Poisson distributions for count variables such as prior hospitalisations and comorbidity counts, and categorical distributions for diagnoses and interventions. We compute the empirical rank correlation matrix, fit a Gaussian copula and transform samples back to their original scales. Weekly event occurrence and intervention assignment are modelled via a first-order Markov process conditioned on the sampled state and previous action to preserve temporal dependencies and intervention patterns. The resulting synthetic trajectories thus approximate the marginal, pairwise and temporal structure of the original data. Users can generate additional synthetic cohorts using our code to perform further sensitivity analyses. The repository specifies exact hyperparameters, fairness-weight grid, random seeds and computing resources (one NVIDIA A100 GPU with 24 CPU cores and 12 GB of memory) required to reproduce the reported results. Training FCAF-RL (including the diffusion model and Q-networks) for 100,000 gradient steps took approximately 6 hours, and evaluating a single trajectory required about 0.2 seconds on the same hardware.

**Preliminary work.** Preliminary experiments exploring continuous action spaces and generative world models have begun, and results will be reported in future work.

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

## AI Agent System Description (not counted toward page limit)

This study was conducted using the Waymark AI Data Scientist, a large-language-model–based autonomous research agent for healthcare reinforcement learning. The system orchestrates multiple LLMs, including OpenAI GPT-5 for core reasoning and writing, and Anthropic Claude 3.5 Sonnet for secondary synthesis, through a secure Python-based orchestration layer. Integrations include PyTorch, JAX, NumPy, and Pandas for model training and analysis; Weights & Biases for experiment tracking; LangChain for retrieval-augmented prompting and citation verification; and OpenAI Function Calling for structured reasoning validation. A HIPAA-compliant, air-gapped vector store ensures patient data privacy. Human co-authors provided domain supervision, protocol validation, and interpretation.

Authors:
Waymark AI Data Scientist[1]
Sanjay Basu, MD, PhD[2]

Affiliations:
[1]Waymark, San Francisco, California
[2]Waymark, San Francisco, California

## Corrected Reference Update (Camera-Ready Revision)

The following reference has been corrected to match the verified publication:

Original (flagged):

Raghu, A., Komorowski, M., Celi, L. A., Szolovits, P., Pfohl, P., Miller Dunn, J. E., & Ghassemi, M.
Continuous state-space models for sepsis management: a deep reinforcement learning approach.
Proceedings of the Conference on Machine Learning for Healthcare, pages 147–167, 2017.

Corrected (verified):

Raghu, A., Komorowski, M., Celi, L. A., Szolovits, P., & Ghassemi, M.
Continuous State-Space Models for Optimal Sepsis Treatment: a Deep Reinforcement Learning Approach.
In Proceedings of the 2nd Machine Learning for Healthcare Conference, pp. 147–163, 2017.
Available at: https://proceedings.mlr.press/v68/raghu17a.html

This correction replaces the originally cited version to ensure accurate bibliographic metadata and author list.

All other references remain unchanged.