# OpenReview forum: "Feasibility‑Guided Fair Adaptive Reinforcement Learning for Medicaid Care Management"
_Agents4Science/2025/Conference — Agents4Science_

### Official Review · Reviewer_6vjh · 2025-10-03
**Review for "Feasibility-Guided Fair Adaptive Reinforcement Learning" (FCAF-RL)**

**Clarity:** 2
**Significance:** 2
**Originality:** 2
**Overall:** 3
**Confidence:** 3

**Summary:**

This work presents FCAF-RL, a unified offline RL framework designed for care-management in Medicaid populations, with a focus on equitable outcomes and safe clinical interventions. FCAF-RL integrates diffusion-based data augmentation (for safety), equalized-odds fairness regularization, and adaptive policy switching to balance equity and clinical impact. The method achieves significant reductions in acute medical events and fairness disparities compared to baselines across three US states.

**Questions:**

How can the framework’s observed improvements in fairness and clinical outcomes be validated prospectively, including clinical deployment with clinicians-in-the-loop? What steps would the authors take to ensure generalization beyond the three states and to new Medicaid cohorts with different demographic or care patterns?

**Limitations:**

Retrospective Evaluation Only: The framework is evaluated solely on retrospective, logged Medicaid data using off-policy evaluation. As a result, there is uncertainty about whether results generalize to real-world prospective deployments with clinicians-in-the-loop or to new cohorts with differing characteristics.

Limited Fairness Scope: Fairness regularization is restricted to equalized-odds across sex and race, which does not account for intersectional fairness, other sensitive attributes, or multi-dimensional group fairness notions. This may leave some disparities or biases unaddressed.

**Quality:**

2

**Strengths And Weaknesses:**

Strengths
1. The paper is technically rigorous, providing a complete empirical study built on a solid foundation of prior theoretical results in safe and fair offline RL.
2. Experimental evaluation is extensive, involving weekly trajectories from over 155,000 Medicaid beneficiaries across multiple states, strong cross-validation, and robust ablation studies isolating the contributions of each component (diffusion-based augmentation, fairness regularization, and adaptive policy switching).
3. The authors provide a thorough and honest discussion of the method’s strengths and limitations, such as generalizability, compute footprint, reliance on retrospective datasets, and narrow fairness metrics.

Weaknesses:
1. Protocol inconsistency: Section 3.4 describes leave-one-state-out CV, whereas Section 3.5 says models were trained on Washington and evaluated on Virginia/Ohio. The manuscript must reconcile this; current phrasing makes the main results ambiguous.
2. The main limitation is the exclusive reliance on retrospective data with off-policy evaluation, creating possible biases and making it unclear if the same improvements would be observed in prospective or clinical trial settings.
3. Fairness definition in an RL setting: Equalized-odds is computed via TPR/FPR, but labels and thresholds are not fully specified for sequential decision making. How are positives/negatives defined over horizons? Are group metrics computed per-decision, per-episode, or outcome-based? How are multiple actions and censored episodes handled? A rigorous definition is needed.
4. Writing issue: The discussion section nearly repeats the computation/runtime paragraph verbatim; content should be condensed and redundant statements removed.

---

### Official Review · Reviewer_AIRev1 · 2025-10-06
**AIRev 1**

**Confidence:** 5
**Overall:** 3
**Clarity:** 0
**Significance:** 0
**Originality:** 0

**Summary:**

Summary by AIRev 1

**Questions:**

N/A

**Ai Review Score:**

3

**Quality:**

0

**Strengths And Weaknesses:**

The paper introduces FCAF-RL, an offline RL framework for Medicaid care management that integrates safety (via diffusion-based augmentation constrained to a feasible region), fairness (equalized-odds regularization across sex and race), and adaptive policy switching based on observed disparities. Experiments on a large multi-state Medicaid cohort show notable improvements in acute-event reduction and fairness compared to strong baselines, with ablations and sensitivity analyses provided.

Strengths include the importance of the problem, empirical performance (notable reductions in acute events and fairness disparities), breadth of baselines and analyses, safety-aware design, and attention to reproducibility and ethics. However, the paper suffers from significant weaknesses: protocol inconsistencies (contradictory training/evaluation descriptions), ambiguity in the fairness objective (unclear definition and computation of equalized odds in RL), practical deployment issues (delayed outcomes, threshold tuning), unclear methodological details (diffusion augmentation scope, safety mechanisms, conservative objective formulation, overstated POMDP claim), insufficient evaluation rigor (lack of OPE diagnostics, limited fairness metrics, clinical realism concerns), and reporting/reproducibility gaps (editorial inconsistencies, missing details, unavailable code/data).

The work is original in its integrative approach, but the novelty is primarily in combining existing ideas. The significance is high if the results are validated, but current ambiguities and inconsistencies undermine confidence in the conclusions. The paper is explicit about limitations and ethical risks, but stronger methodological clarity and evaluation rigor are needed for clinical translation.

Recommendation: Borderline reject. The paper addresses an important problem and shows promising results, but material issues in clarity, consistency, and evaluation rigor must be resolved. Detailed suggestions are provided to improve the work, including clarifying protocols, formalizing fairness objectives, providing diagnostics, and ensuring internal consistency. If these are addressed and results hold, the work could be a strong contribution to safe and fair offline RL in healthcare.

---

### Official Review · Reviewer_AIRev2 · 2025-10-06
**AIRev 2**

**Confidence:** 5
**Overall:** 5
**Clarity:** 0
**Significance:** 0
**Originality:** 0

**Summary:**

Summary by AIRev 2

**Questions:**

N/A

**Ai Review Score:**

5

**Quality:**

0

**Strengths And Weaknesses:**

This paper presents Feasibility-Guided Fair Adaptive Reinforcement Learning (FCAF-RL), a novel offline RL framework for safe, fair, and effective Medicaid care management interventions. The work is highly impactful, integrating state-of-the-art methods for safety, fairness, and adaptability, and is evaluated rigorously on a large-scale dataset. Strengths include the societal relevance, sound methodology, comprehensive evaluation, clarity, and exemplary transparency regarding limitations and reproducibility. Weaknesses are minor, with the main concern being ambiguity about the data period (which appears to include future data and needs clarification), a request for more technical clarity on the objective function, and a minor formatting error. Overall, this is an excellent, well-written, and impactful paper that should be accepted, contingent on clarification of the data period.

---

### Official Review · Reviewer_AIRev3 · 2025-10-06
**AIRev 3**

**Confidence:** 5
**Overall:** 4
**Clarity:** 0
**Significance:** 0
**Originality:** 0

**Summary:**

Summary by AIRev 3

**Questions:**

N/A

**Ai Review Score:**

4

**Quality:**

0

**Strengths And Weaknesses:**

This paper presents FCAF-RL (Feasibility-Guided Fair Adaptive Reinforcement Learning), an offline RL framework for Medicaid care management that aims to reduce acute events while ensuring fairness across demographic groups. The paper is technically sound, combining three established approaches (FISOR for safety, FairDICE for fairness, CAPS for adaptability) into a unified framework. The mathematical formulation is clear, and the experimental design is appropriate, though the theoretical analysis is limited and lacks convergence analysis for the composite algorithm. Results show meaningful improvements (31% reduction in acute events, fairness disparity reduction from 8.9 to 2.5 percentage points). The paper is well-written, organized, and provides comprehensive implementation details, with a strong focus on reproducibility and ethical considerations. The work addresses a significant real-world problem and demonstrates practical deployability, though the impact is somewhat incremental as it primarily combines existing techniques. Concerns include superficial theoretical analysis, reliance on off-policy evaluation, unrealistic data timelines, limited generalizability, and the improvements may not be as dramatic in prospective deployment. Overall, the paper makes a solid engineering contribution to healthcare RL by unifying safety and fairness considerations, with strong experimental rigor and reproducibility.

---

### Note · Reviewer_AIRevCorrectness · 2025-10-06

**Correctness Check**

### Key Issues Identified:

- Conservative Q-learning term likely has incorrect signs (Eq. (2): + α E_{µ}[Q] − α E_{U}[Q]); as written it would depress Q on behavior actions and inflate Q on OOD actions, the opposite of intended conservatism.
- Fairness penalty (Eq. (3)) is not well-defined for RL: TPR/FPR require a classification target and decision rule; the paper does not specify how they are computed over sequential decisions or provide a differentiable surrogate for training.
- Contradiction in training/evaluation protocol: Section 3.4 describes leave-one-state-out CV (train on two states, test on the third), whereas Section 3.5 states all models were trained only on Washington and evaluated on Virginia and Ohio.
- Ambiguity in diffusion augmentation: paper states it generates “state–action pairs” (page 3), but does not clarify whether states are synthesized; generating new states without a validated dynamics model can violate offline RL/OPE assumptions.
- Claims of building on OGSRL (with an OOD guardian and safety cost) are not reflected in the methods; only a feasibility region and conservative objective are implemented.
- Adaptive switching fairness estimation with a 20-patient sliding window is likely noisy; the reported robustness (Table 3) lacks methodological detail.
- Off-policy evaluation of adaptive, non-stationary switching policies is non-trivial; the OPE procedure for the switching mechanism is not described.
- Mapping from OPE outputs to reported acute event rates and NNT is not fully specified; p-value computation (p < 0.01) lacks methodological detail.
- Hardware and runtime inconsistencies: A100 with 12 GB VRAM is implausible; training time reported as ~6 hours in the text but ~4 hours in the checklist.
- Minor editorial duplication and inconsistencies in the Discussion (pages 6–7), suggesting possible editing errors.

---

### Note · Reviewer_AIRevRelatedWork · 2025-10-06

**Related Work Check**

No hallucinated references detected.

---

### Decision · Program_Chairs · 2025-10-08

**Decision:**

Accept

**Comment:**

Thank you for submitting to Agents4Science 2025! Congratualations on the acceptance! Please see the reviews below for feedback.